# Phytaspase Is Capable of Detaching the Endoplasmic Reticulum Retrieval Signal from Tobacco Calreticulin-3

**DOI:** 10.3390/ijms242216527

**Published:** 2023-11-20

**Authors:** Anastasia D. Teplova, Artemii A. Pigidanov, Marina V. Serebryakova, Sergei A. Golyshev, Raisa A. Galiullina, Nina V. Chichkova, Andrey B. Vartapetian

**Affiliations:** 1Faculty of Bioengineering and Bioinformatics, Lomonosov Moscow State University, Moscow 119991, Russia; anastasia_teplova@mail.ru (A.D.T.); artemiy.pigidanov@mail.ru (A.A.P.); 2Belozersky Institute of Physico-Chemical Biology, Lomonosov Moscow State University, Moscow 119991, Russia; mserebr@mail.ru (M.V.S.); sergei.a.golyshev@gmail.com (S.A.G.); raisa-galiullina@rambler.ru (R.A.G.); chic@belozersky.msu.ru (N.V.C.)

**Keywords:** calreticulin-3, endoplasmic reticulum, phytaspase, plant cell, protein trafficking, proteolytic processing, retrieval signal

## Abstract

Soluble chaperones residing in the endoplasmic reticulum (ER) play vitally important roles in folding and quality control of newly synthesized proteins that transiently pass through the ER en route to their final destinations. These soluble residents of the ER are themselves endowed with an ER retrieval signal that enables the cell to bring the escaped residents back from the Golgi. Here, by using purified proteins, we showed that *Nicotiana tabacum* phytaspase, a plant aspartate-specific protease, introduces two breaks at the C-terminus of the *N. tabacum* ER resident calreticulin-3. These cleavages resulted in removal of either a dipeptide or a hexapeptide from the C-terminus of calreticulin-3 encompassing part or all of the ER retrieval signal. Consistently, expression of the calreticulin-3 derivative mimicking the phytaspase cleavage product in *Nicotiana benthamiana* cells demonstrated loss of the ER accumulation of the protein. Notably, upon its escape from the ER, calreticulin-3 was further processed by an unknown protease(s) to generate the free N-terminal (N) domain of calreticulin-3, which was ultimately secreted into the apoplast. Our study thus identified a specific proteolytic enzyme capable of precise detachment of the ER retrieval signal from a plant ER resident protein, with implications for the further fate of the escaped resident.

## 1. Introduction

Proteins are targeted to various specific locations within or outside the cell with the aid of localization signals—usually rather short amino acid sequences—within protein molecules. This notion holds true for soluble proteins which are ‘residents’ of the endoplasmic reticulum (ER). To fulfill their tasks related to protein folding and quality control within the ER, such proteins are usually equipped with two kinds of localization signals, each with a terminal location within the protein molecule. The first one is the N-terminal signal peptide (SP) that allows targeting of the newly synthesized protein to the ER [1,2,3] and which is cleaved off in the process of protein translocation by signal peptidase [4,5]. The second one is the C-terminal ‘ER retrieval signal’ (usually KDEL in animals or HDEL in plants and yeast) [6,7,8], which is used to bring back these soluble chaperones when they move from the ER toward the Golgi apparatus, together with thousands of other proteins that are delivered through the ER to their final destinations. The ER retrieval signal within the escaped residents is recognized by the receptor ERD2 located at the cis-Golgi [9,10,11,12,13] to mediate retrograde transportation of these proteins to the ER.

In contrast to the cleavable SP, the ER retrieval signal is thought to be non-cleavable, or at least no proteolytic enzyme in any kingdom of life is known that is able to specifically detach the ER retrieval signal from the protein. In this regard, it is surprising that for decades, soluble chaperones of the ER have also been observed at other locations both within and outside the cell (see, for example, [14,15,16,17,18]). Furthermore, experimental evidence suggests that these ‘escaped residents’ fulfill important tasks at these non-canonical locations [19,20,21,22]. An essential unresolved question is how these ER residents managed to escape despite the existence of an elaborate retrieval mechanism.

Here, we present in vitro evidence that phytaspase (plant aspartate-specific protease), a member of the plant subtilisin-like protease (subtilase) family, is capable of precise detachment of the ER retrieval signal from *N. tabacum* calreticulin-3 (*Nt*CRT3), an ER resident protein. Two features distinguish phytaspases from other members of the extensive family of plant subtilases. First, phytaspases possess rarely occurring aspartate cleavage specificity. Like animal apoptotic proteases, namely caspases [23,24,25], phytaspases hydrolyze the polypeptide chain after an Asp residue preceded by a characteristic though degenerate three amino acid-long motif [26,27,28]. This type of substrate recognition makes phytaspases highly selective enzymes that introduce one or two breaks in a limited number of protein substrates. Phytaspases were initially identified as plant proteases involved in the execution of programmed cell death induced by various biotic and abiotic insults [26,29,30,31], but they were subsequently shown to also be instrumental in specific processing of precursors of the plant peptide hormones systemin and phytosulfokine to generate biologically active peptides [32,33].

Another peculiar feature of phytaspases is their dynamic localization. In healthy plant cells, the phytaspase precursor equipped with an SP is targeted to the ER, is constitutively and autocatalytically processed and thus activated on its way out of the cell, and the mature enzyme is then secreted into the apoplast [26]. In response to a number of biotic and abiotic stresses, phytaspase is internalized back into the cell by means of clathrin-mediated endocytosis to participate in the execution of programmed cell death [26,34,35].

By using the BioID approach, the in vivo interaction of *N. tabacum* phytaspase (*Nt*Phyt) with *N. benthamiana* CRT3 and several other ER resident chaperones was documented [36]. Notably, CRT3 from *Nt*Phyt-overproducing plant cells displayed a slightly increased electrophoretic mobility (relative to CRT3 from the control cells), implying the existence of some kind of post-translational modification [36]. Since phytaspase is a protease, such a modification could consist of phytaspase-mediated processing of CRT3. Here, we assess this assumption using purified proteins. We demonstrate that *Nt*CRT3 is not only the partner but also the substrate of *Nt*Phyt. We also show that the *Nt*CRT3 derivative mimicking the phytaspase cleavage product escapes from the ER and is further processed to generate the N-domain of CRT3, with the latter being secreted into the apoplast.

## 2. Results

### 2.1. Phytaspase-Mediated Truncation of N. tabacum CRT3: Evidence with Purified Proteins

We assessed the hypothesis that CRT3 is a substrate of tobacco phytaspase using purified proteins. *N. tabacum* CRT3 bearing a His tag instead of the native signal peptide was produced in *E. coli* cells and purified by means of Ni-NTA agarose affinity chromatography. *N. tabacum* phytaspase (*Nt*Phyt) with a C-terminal His tag was transiently overproduced in *N. benthamiana* leaves by agroinfiltration and isolated from the apoplast by affinity chromatography. Treatment of the recombinant CTR3 with *Nt*Phyt resulted in a small yet clearly detectable shift in the electrophoretic mobility of CRT3 (Figure 1A) similar to that observed previously with extracts from *Nt*Phyt-producing versus non-producing *N. benthamiana* leaves in our BioID experiments [36].

To verify that the truncation of CRT3 occurred due to phytaspase-specific proteolytic activity, purified recombinant *Nt*Phyt was pretreated with either Ac-VEID-CHO or Ac-DEVD-CHO peptide aldehyde prior to addition of the enzyme to CRT3. The former is the known inhibitor of *Nt*Phyt [26], whereas the latter is a specific inhibitor of animal caspases-3 that is known to not interfere with the *Nt*Phyt proteolytic activity [26,29]. As evidenced by Figure 1B, truncation of CRT3 was completely abolished by the *Nt*Phyt-specific inhibitor, while pretreatment with the control peptide aldehyde had no effect.

These results suggest that *Nt*Phyt is capable of cleaving a short peptide from one of the CRT3 termini.

### 2.2. Identification of the NtPhyt Cleavage Site(s) in CRT3

To identify the *Nt*Phyt cleavage sites, the CRT3 bands from the phytaspase-treated and non-treated samples shown in Figure 1A were excised and treated with LysC protease, and the resultant peptides were characterized by MALDI TOF mass spectrometry analyses. While the majority of the peptides, including the N-terminal, of both the *Nt*Phyt-treated and non-treated CRT3 displayed identical peaks in the mass spectra (Appendix A), the C-termini turned out to be different. In the untreated sample, the intact C-terminal semi-LysC peptide with the predicted *m*/*z* 2064.9 was readily identified, whereas in the *Nt*Phyt-treated sample, a shorter semi-LysC peptide with *m*/*z* 1292.6 was observed instead (Figure 1C; for the original MS data, see Appendix A). The identity of the characteristic peptides was confirmed by MS/MS fragmentation (Appendix A). These findings indicate that *Nt*Phyt cleaved off the C-terminal hexapeptide of CRT3, and the hydrolysis occurred in accord with the *Nt*Phyt specificity (that is, after an aspartate residue D) in the DYMD^420^ motif.

To further substantiate this conclusion, we substituted D^420^ with E (glutamate) in CRT3 with the expectation of obtaining a phytaspase-resistant version of CRT3. (*Nt*Phyt is a strictly D-specific protease.) Upon isolation from the bacterial overproducing strains, the wild-type and D420E mutant proteins were treated with *Nt*Phyt, and the reaction products were compared using SDS gel-electrophoresis. Indeed, the D420E mutation abolished the characteristic mobility shift observed with the wild-type CRT3 (Figure 2A). However, the CRT3 band in the D420E phytaspase-treated sample still appeared to migrate somewhat faster than that in the non-treated counterpart (Figure 2A, compare lanes 3 and 4). To clarify this issue, mass spectrometry analysis of the LysC peptides generated from phytaspase-treated and untreated CRT3 D420E bands was performed. The intact C-terminal semi-LysC peptide of CRT3 D420E encompassing the D420E mutation (*m*/*z* 2078.9) was identified in the untreated sample (Figure 2B). However, in the *Nt*Phyt-treated sample, a shorter semi-LysC peptide (*m*/*z* 1836.8) was observed instead, corresponding to elimination of just two C-terminal amino acid residues of CRT3 D420E (Figure 2B; for the complete spectra of peptides from both samples, see Appendix A). The cleavage therefore occurred after the DYHD^424^ motif and again with the characteristic *Nt*Phyt specificity.

We therefore concluded that there are two rather similar *Nt*Phyt cleavage sites (DYMD^420^ and DYHD^424^) positioned in extremely close proximity to the CRT3 C-terminus. The *Nt*Phyt-mediated cleavage appeared to proceed more efficiently at D^420^ than at D^424^, as we failed to detect the elimination of only the two C-terminal residues unless the D^420^ was mutated. Also consistent with this interpretation, the cleavage of the mutant protein at D^424^ was incomplete (Appendix A), in contrast to complete hydrolysis of the wild-type protein at D^420^ (Figure 1A and Figure 2A) under similar conditions of hydrolysis.

### 2.3. The Effect of NtCRT3 Truncation on the Protein Localization and Integrity

CRT3, like other soluble residents of the ER, contains an ‘ER retrieval signal’ represented by the HDEL sequence at the very C-terminus of the molecule (Figure 3A). This signal allows the cell to return CRT3 that had escaped from the ER back from the Golgi, a feature actually making this soluble protein an ER resident. Phytaspase-mediated cleavage to detach either half of or the entire ER retrieval signal would thus be predicted to alter at least the subcellular localization of CRT3. To test this assumption, we constructed two fluorescent *Nt*CRT3 derivatives. The EGFP_CRT3_wt construct contained the full-length CRT3 protein with the EGFP tag inserted next to the N-terminal signal peptide in order to not interfere with the terminal localization signals in CRT3. The EGFP_CRT3_ΔC6 construct had a similarly positioned EGFP tag, yet it lacked six C-terminal amino acid residues of CRT3 encompassing the ER retrieval signal, thus mimicking the phytaspase cleavage product of CRT3 (see Figure 3A for a schematic representation). Both proteins were transiently produced in the *N. benthamiana* leaves by agroinfiltration, and their localization was compared using confocal fluorescence microscopy. The EGFP_CRT3_wt protein displayed characteristic ER localization, as expected (Figure 3B). The production of EGFP_CRT3_ΔC6 resulted in a much lower fluorescence intensity (relative to the wt construct) in the infiltrated leaves, with fluorescence shifting from the ER to the cell’s periphery (Figure 3C).

To characterize the behavior of these proteins in greater detail, we separated the proteins from the infiltrated leaves into intra- and extracellular (apoplastic) fractions and analyzed them through Western blotting using an anti-EGFP antibody. The EGFP_CRT3_wt protein (calculated mass of ~80 kDa) was largely intact and detected in the intracellular fraction (Figure 3D, left side). Contrary to that, the EGFP_CRT3_ΔC6 protein was degraded, forming a ~30 kDa product (Figure 3D, right), which is approximately the size of free EGFP. This product was observed in the extracellular fraction (Figure 3D). To learn what happened to the CRT3 moiety of the fusion proteins, analogous Western blots were analyzed using anti-CRT3 antibodies. In accord with the previous analysis, the EGFP_CRT3_wt protein was found to be predominantly intact, although some degradation products were also observed, and it was localized intracellularly (Figure 4A). For the EGFP_CRT3_ΔC6 protein, although a minor amount of the intact (~80 kDa) protein was detected on the overexposed blot (Figure 4B, lane 2), the majority of this protein was degraded to produce a ~26 kDa protein fragment, the most intense band recognized by anti-CRT3 antibodies. Interestingly, this CRT3 fragment was found in the extracellular fraction as well (Figure 4B, lane 3). Also of note is that the endogenous CRT3 observed in the ‘vector only’ lane (Figure 4B, lane 1) was fairly intact.

The above results suggest that *Nt*CRT3 lacking the ER retrieval signal is further processed within the plant cell, and a protein fragment is secreted into the apoplast. To obtain insights into where such additional processing could occur, the effect of brefeldin A (BFA), a known inhibitor of protein transport from the ER to the Golgi complex [37,38], on EGFP_CRT3_ΔC6’s behavior was studied. Treatment of the EGFP_CRT3_ΔC6-producing *N. benthamiana* leaves with BFA markedly impaired the formation of the 26 kDa CRT3 fragment and abolished its secretion, with a concomitant increase in the amount of the full-length protein (Figure 4B, lanes 4 and 5). These results are consistent with fragmentation of EGFP_CRT3_ΔC6 occurring within the plant cell upon protein release from the ER.

Finally, to characterize the secretion-competent 26 kDa CRT3 fragment, the EGFP_CRT3_ΔC6 construct was expressed in *N. benthamiana* leaves in the presence of p19 RNA-silencing suppressor to maximize production. The rather abundant 26 kDa CRT3 fragment thus formed was isolated from the apoplastic fraction (Figure 4C) and, after trypsin or Glu-C digestion, identified by mass spectrometry analysis as the N-terminal domain of CRT3, in agreement with its recognition by anti-CRT3 antibodies (see Figure 5). Judging by the pattern of peptides identified, the CRT3 cleavage both in the vicinity of the N-P domain border and at the N-terminus of the N domain by an unknown protease(s) was required to generate this CRT3 fragment.

## 3. Discussion

In this study, we provide in vitro evidence that *Nt*Phyt efficiently and precisely removes the ER retrieval signal of *Nt*CRT3 by introducing two breaks at the very C-terminus of the *Nt*CRT3 polypeptide. We also demonstrated that the *Nt*CRT3 derivative mimicking the phytaspase cleavage product loses the capacity for accumulation in the ER and is further processed by an unknown protease(s), and the stable N domain of *Nt*CRT3 thus formed is ultimately secreted into the apoplast. From the ‘functional’ point of view, the need for two similar and closely spaced phytaspase cleavage sites at the C-terminus of the *Nt*CRT3 molecule is not straightforward. It would seem that one would be sufficient to destroy the ER retrieval signal completely. On the other hand, the C-terminal decapeptides of, for example, *Arabidopsis thaliana* and *N. tabacum* CRT3 are identical in terms of amino acid sequence, suggesting that such duplication is functionally important.

Another question is clearly of greater importance. Perhaps unsurprisingly, phytaspase-mediated trimming of CRT3 seems to not occur (or at least is inefficient) in *N. benthamiana* cells under normal conditions. Otherwise, the ER would lose its highly important chaperone [12,39,40,41,42,43]. It would therefore be essential to learn whether stress-inducing or some other conditions could trigger phytaspase-mediated cleavage of the ER retrieval signal from *Nt*CRT3. An interaction of phytaspase with CRT3 was documented in vivo using the BioID approach [28]. In this regard, consideration of the complex trafficking pattern of phytaspase could suggest mechanisms for triggering the relocalization of *Nt*CRT3.

In principle, there are two opportunities for phytaspase to encounter CRT3: either during the export of phytaspase out of the cell (for the newly synthesized enzyme) or upon the stress-induced retrograde transportation (for the preexisting phytaspase). For the first scenario, a caveat is that plant subtilases are believed to not acquire activity until they reach the Golgi apparatus [44,45,46], and proteolytically active phytaspase should therefore not be present in the ER. On the other hand, the ER retrieval signal endows soluble residents of the ER with the ability to shuttle between the ER and the Golgi [13]. It thus can be envisaged that proteolytic processing of CRT3 by phytaspase could occur in the Golgi, where phytaspase is already proteolytically competent, thus disabling CRT3 retrieval back to the ER. This regulatory option, if it turns out to be true, is achievable due to the difference (possibly, the advantage) of the retrieval mechanism for accumulation of the soluble residents in the ER over a simple retention mechanism.

The second scenario would predict that phytaspase, upon its stress-induced re-import, can process CRT3. The internalized phytaspase is known to retain its proteolytic activity [35], but the precise trafficking routes of the enzyme upon re-entering the cell are not known. Perhaps some kind of a precedent could be provided by the plant protein toxin ricin and related toxins, which can reach the Golgi apparatus and the ER after being endocytosed [47,48,49,50]. Deciphering the phytaspase trafficking pathways in stressed plant cells may thus contribute to understanding of the phytaspase-CRT3 interaction.

The formation and release into the extracellular space of the N domain derived from CRT3_ΔC6 also deserves some comments. If phytaspase-mediated processing of CRT3 triggers formation of such a CRT3 fragment in vivo, then a function of the secreted N domain to serve as a signaling molecule can be envisioned. Notably, the N domain of human calreticulin found in culture supernatants of human cells and named vasostatin was demonstrated to inhibit the proliferation of endothelial cells and to suppress angiogenesis in vivo [51].

In conclusion, our study seems to provide an illustration related to the long considered problem of how soluble residents of the ER equipped with the ER retrieval signal can escape to new locations [16]. The presence of a highly specific dedicated protease (phytaspase) capable of precise detachment of the retrieval signal from the ER resident protein offers plants an intriguing opportunity.

## 4. Materials and Methods

### 4.1. Plant Growth Conditions

*N. benthamiana* plants were grown at 25 °C in soil in a controlled environment under a 16/8 h day/night cycle. For transient protein production, *Agrobacterium tumefaciens* C58C1 cells transformed with the respective plasmid were infiltrated into leaves of 6 week-old plants using a blunt syringe. Typically, the leaves were harvested 2 days post infiltration (dpi).

### 4.2. Plasmid Construction

Construction of the pSL1180_NtCRT3 and pET_His_LF CRT3 plasmids encoding full-length *Nt*CRT3 or *Nt*CRT3 with the SP substituted with the His tag, respectively, was described previously [36]. The D420E *Nt*CRT3 mutant was created by site-directed mutagenesis via PCR on the pET_His_LF CRT3 plasmid using CRT3_Pst700_dir and CRT3_D420E_Sac_rev primers (Appendix A), with the subsequent substitution of the original *Pst*I-*Sac*I DNA fragment of the pET_His_LF CRT3 plasmid with the mutant one.

For the in planta production of EGFP_CRT3_wt, the SP-encoding DNA fragment (circa 110 bp) was amplified on the pSL1180_NtCRT3 using CRT_Kpn_Nco_dir and CRT_SP_Sal_rev primers, inserted into the *Sma*I site of the pUC19 vector, and further transferred as the *Bam*HI-*Sal*I DNA fragment (circa 110 bp) into the pBluescript II vector to generate the pBS_SP plasmid. The EGFP gene (circa 750 bp) was amplified on the pEGFP-N1 vector (Clontech, Palo Alto, CA, USA) using EGFP_Sal_dir and EGFP_Apa_rev primers and inserted downstream of and in frame with the SP-encoding sequence between the *Sal*I and *Apa*I sites of the pBS_SP plasmid to generate the pBS_SP_EGFP plasmid. Finally, the SP-EGFP-encoding *Nco*I-*Apa*I DNA fragment (circa 850 bp) from the latter plasmid and the *Apa*I-*Sac*I DNA fragment (circa 1200 bp) from the pET_His_LF CRT3 plasmid encoding CRT3 without the SP were ligated between the *Nco*I and *Sac*I sites of the pCambia1300EX binary vector [35] to generate the pCambia_SP_EGFP_CRT3_wt plasmid. For production of the EGFP_CRT3_ΔC6 protein, the mutated CRT3-encoding DNA fragment (circa 1200 bp) was obtained by PCR on the pET_His_LF CRT3 plasmid with CRT_LF_Apa_Nde_dir and CRT_1-420_Sac_rev primers. Substitution of the wild-type CRT3-encoding *Apa*I-*Sac*I DNA fragment within the pCambia_SP_EGFP_CRT3_wt plasmid for the mutant one generated the pCambia_SP_EGFP_CRT3_ΔC6 plasmid.

To create the *Nt*Phyt-His fusion, the *Nt*Phyt cDNA from the previously described *Nt*Phyt-GST construct [26] within the pLEX7000 backbone [32] was PCR amplified using pLH_seq_dir and NtPhyt-His_rev primers and inserted between the *Xho*I and *Sac*I sites of the pLEX7000 expression vector downstream of the dual CaMV 35S promoter to generate the pLEX_NtPhyt_His plasmid.

The identities of all constructs were confirmed using sequence analysis.

### 4.3. Production, Purification, and Treatment of the His-Tagged NtCRT3 with NtPhyt

Production of the His-tagged *Nt*CRT3 in *E. coli* BL21(DE3) cells and purification of the recombinant protein using metal chelate affinity chromatography on Ni-NTA agarose (Qiagen, Hilden, Germany) was performed as described previously [36]. The protein was eluted from the beads with 100 mM EDTA in B1 buffer (50 mM MES, pH 5.7, 50 mM NaCl, 2mM DTT, 0.1% Tween 20, and 5% glycerol) and stored at −28 °C. The purification procedure was also performed with the lysate of *E. coli* BL21 (DE3) cells transformed with the empty pET28a(+) vector (Novagen, Madison, WI, USA) to use the obtained eluate as a control.

Purified His-CRT3 (wt or D420E) in 0.5–1 μg aliquots was mixed with 15 ng of *Nt*Phyt-His (see Section 4.5 below for *Nt*Phyt-His isolation) in the total volume of 40 μL of B1 buffer containing 500 mM NaCl, chymostatin (6 μg/mL), AEBSF (25 μg/mL), and E-64 (2 μg/mL) proteinase inhibitors. The reaction mixture was incubated at 28 °C for 1 h. To terminate the reaction, 10 μL of 5xSample buffer (250 mM Tris-HCl, pH 6.8, 5 mM EDTA, 10% SDS, 30% glycerol, and 10 mM DTT) was added to the reaction tube, and the sample was immediately heated for 5 min at 98 °C. As a control, an equal aliquot of His-CRT3 without the addition of *Nt*Phyt was incubated identically. The protein samples were analyzed using 10% SDS-PAGE. The gel was run at a low voltage (150 V) until the 35 kDa protein marker reached the edge of the gel, and it was stained with Coomassi blue R-250.

To test the effect of peptide aldehyde inhibitors, *Nt*Phyt-His (in approximately 6 ng aliquots) was preincubated with 200 μM of Ac-VEID-CHO or Ac-DEVD-CHO (from stock solutions in DMSO) or with the equivalent amount of DMSO (mock control) in the total volume of 20 μL of B1 buffer containing 500 mM NaCl at 28 °C for 30 min. Then, 0.5–1 μg of His-CRT3 in 20 μL of the same buffer was added to each preincubated *Nt*Phyt aliquot, and the samples were processed as described above.

### 4.4. MS Analyses

Pieces of about 2 mm^3^ of the Coomassie-stained protein-containing gel were destained twice with a 20 mM NH_4_HCO_3_ (pH 7.5, 40% aqueous acetonitrile) solution and dehydrated with 200 μL of 100% acetonitrile and rehydrated with 5 μL of the digestion solution containing 15 µg/mL sequencing-grade Lys-C or Glu-C protease or trypsin (Promega, Madison, WI, USA) in a 20 mM NH_4_HCO_3_ (pH 7.5) aqueous solution. Digestion was carried out at 37 °C for 3 h. The resulting peptides were extracted with 7 μL of a 0.5% TFA, 50% acetonitrile solution. A 1 μL aliquot of in-gel digest extract was mixed with 0.5 μL of a 2,5-dihydroxybenzoic acid solution (40 mg/mL in 30% acetonitrile, 0.5% TFA) and left to dry on the stainless steel target plate. MALDI-TOF MS analysis was performed on an UltrafleXetreme MALDI-TOF-TOF mass spectrometer (Bruker Daltonik, Bremen, Germany). The MH^+^ molecular ions were measured in a reflector mode, and the accuracy of monoisotopic mass peak measurement was within 100 ppm. Spectra of fragmentation were obtained in LIFT mode, and the accuracy of the daughter ion measurement was within a 1 Da range. The mass spectra were processed with FlexAnalysis 3.2 software (Bruker Daltonik, Bremen, Germany). Protein identification was carried out through an MS + MS/MS ion search with the use of Mascot software version 2.3.02 (Matrix Science, London, UK) through the Home protein database. One missed cleavage, Met oxidation, and Cys-propionamide were permitted. The mass spectra of the fragmentation of C-terminal peptides were labeled manually.

### 4.5. Transient Expression in N. benthamiana and Protein Fractionation

The obtained binary plasmids encoding EGFP-*Nt*CRT3 derivatives were introduced into *A. tumefaciens* C58C1 cells, and transformed agrobacteria were infiltrated into *N. benthamiana* leaves. Agrobacteria carrying the empty vector were used as a control. Where indicated, leaves were infiltrated with BFA (15 μg/mL) 1 day after agroinfiltration of the EGFP_CRT3 construct and harvested 24 h later. At 2 dpi, the leaves were examined using confocal fluorescence microscopy. In parallel, the leaves were subject to protein fractionation essentially as described previously [35] with some modifications. Briefly, the leaf discs were weighed and vacuum infiltrated with water. Apoplastic washes were obtained through low-speed (2000× *g*) centrifugation at 4 °C for 10 min, with the liquid collecting directly into 5×Sample Buffer placed at the bottom of a centrifuge tube. After the separation, the residual leaf material was frozen in liquid nitrogen and disrupted in Minilys homogenizer (Bertin Instruments, Montigny-le-Bretonneux, France) using 1.6 mm ceramic beads in three 10 s bursts. After the addition of the 5 × Sample Buffer, the homogenates were boiled for 5 min, debris was eliminated through 10 min of centrifugation at 10,000× *g*, and the protein samples were fractionated by SDS-polyacrylamide gel electrophoresis. The separated proteins were electrophoretically transferred onto a polyvinylidene fluoride (PVDF) membrane. Western blot analyses were performed using mouse monoclonal anti-EGFP 3A9 antibody [35] and rabbit polyclonal anti-*Nt*CRT3 antibodies. The rabbit anti-*Nt*CRT3 antibodies were raised against the synthetic NtCRT3 peptide GSMYTDNDILPPRKIK by GenScript USA. Chemiluminescence detection was performed with ECL Western Lightning Plus reagent (PerkinElmer, Waltham, MA, USA) using the ChemiDoc Imaging System (Bio-Rad Laboratories, Hercules, CA, USA).

To enhance the protein yield when desirable (e.g., for production of *Nt*Phyt-His or for the experiment shown in Figure 4C), transformed agrobacteria were mixed with an equal amount of *Agrobacterium* cells bearing the p19 suppressor of silencing prior to infiltration. The leaves were harvested at 3 dpi.

To isolate the proteolytically active *Nt*Phyt-His, apoplastic washes were obtained from approximately 3 g of leaves by vacuum infiltration with MES100 buffer (20 mM MES, pH 5.7, and 100 mM NaCl) as described in [35]. Apoplastic liquid was then diluted 10 fold with 50 mM Na-phosphate buffer (pH 8.0), supplemented with aprotinin (2 μg/mL), leupeptin (6 μg/mL), AEBSF (25 μg/mL), and chymostatin (6 μg/mL) proteinase inhibitors, and applied onto Ni-NTA agarose (~200 μL suspension) pre-washed with the same buffer. After incubation with rotation at 4 °C for 1.5 h, the beads were separated by centrifugation at 2000× *g* for 10 min and then washed with Na-phosphate buffer (pH 8.0) containing 1 M NaCl and 10% glycerol (10 × 1 mL) and finally with 1 mL of Na-phosphate buffer. *Nt*Phyt-His was then eluted from the beads with 20 mM MES, pH 6.5, containing 100 mM EDTA (3 x 100 μL). The eluate was transferred into B1 buffer containing 50 mM NaCl or into 0.1 M HEPES (pH 8.0) buffer using a centrifugal concentrator with a molecular weight cut-off of 50,000 (Amicon Ultra-4) and stored at −28 °C. Phytaspase proteolytic activity was evaluated using fluorogenic peptide substrate Ac-VEID-AFC as described in [35].

### 4.6. Confocal Fluorescence Microscopy

Confocal microscopy was performed using a Nikon C2plus confocal microscope on a Nikon Eclipse Ti inverted stand (Nikon, Tokyo, Japan) equipped with a VC PlanApo 60x NA1.2 water immersion lens. A 486 nm laser was used for excitation of EGFP. Stacks of optical sections of the apical half of the epidermal cells were acquired at a z-step of 0.8 μm. The obtained data are presented as maximum intensity projection images.

## Figures and Tables

**Figure 1 ijms-24-16527-f001:**
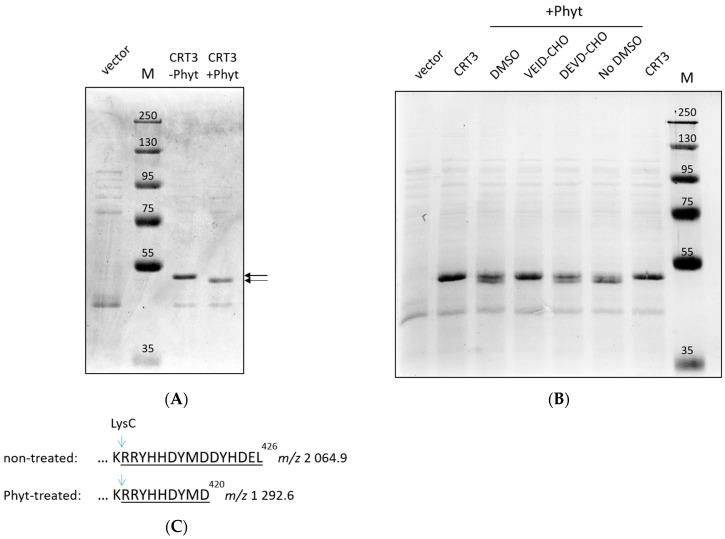
*Nt*CRT3 is the substrate of *Nt*Phyt. (**A**) Treatment of isolated recombinant His-calreticulin-3 (CRT) with phytaspase (Phyt) caused a small increase in the electrophoretic mobility of CRT3 (arrows). Vector = protein fraction obtained upon Ni-NTA agarose affinity chromatography from *E. coli* cells transformed with empty vector. M = MW protein markers. (**B**) Hydrolysis was suppressed by Ac-VEID-CHO but not by Ac-DEVD-CHO. Both inhibitors were used at a concentration of 200 μM. In (**A**,**B**), reaction products were fractionated by 10% SDS-gel electrophoresis and stained with Coomassie blue. (**C**) Schematic representation of the phytaspase cleavage product of CRT3. Mass spectrometry (MS) analysis revealed characteristic semi-LysC peptides with *m*/*z* 2064.9 and 1292.6 derived from untreated and Phyt-treated CRT3 bands, respectively (arrows in (**A**)).

**Figure 2 ijms-24-16527-f002:**
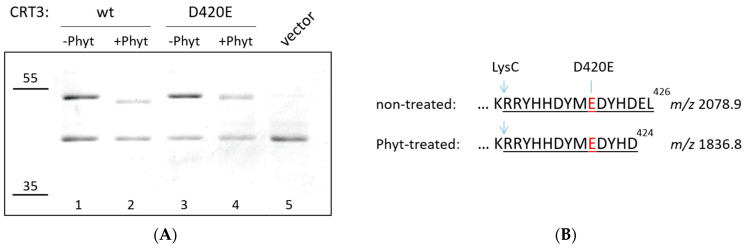
Mutating D^420^ in CRT3 revealed the second Phyt cleavage site located within the ER retrieval signal (HDEL^426^). (**A**) Recombinant N-terminally His-tagged wild-type (wt) and mutant (D420E) CRT3 were isolated from *E. coli* cells and treated with *Nt*Phyt (+Phyt) or left untreated (−Phyt), and reaction products were fractionated by 10% SDS-gel electrophoresis and stained with Coomassie blue. Note a slight mobility shift of the D420E mutant protein caused by *Nt*Phyt treatment. Vector = an analogous protein fraction from *E. coli* cells transformed with empty vector. The positions of the MW protein markers are indicated on the left. (**B**) Schematic representation of the phytaspase cleavage product of the CRT3 D420E mutant. The mutation is shown in red. Characteristic semi-LysC peptides with *m*/*z* 2078.9 (untreated sample) and 1836.8 (Phyt-treated) were identified using MS analyses.

**Figure 3 ijms-24-16527-f003:**
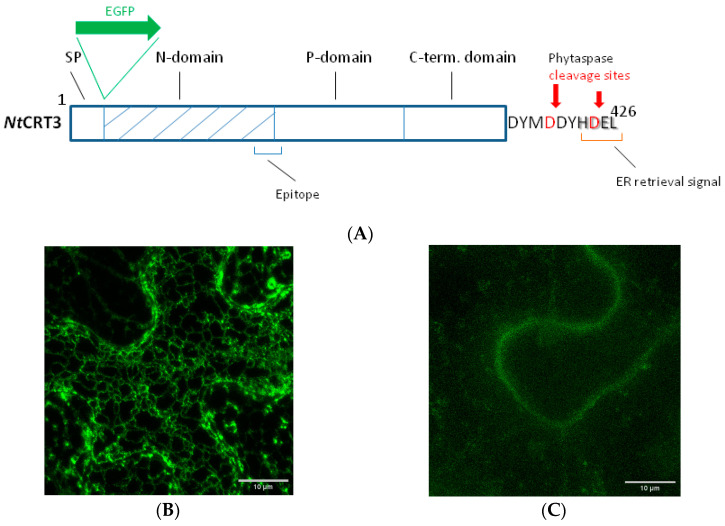
EGFP-CRT3 derivative mimicking the major phytaspase cleavage product loses the ER accumulation and is further processed. (**A**) Schematic representation of *Nt*CRT3 structure and features addressed in the current study. SP = signal peptide. Position of the insertion of the EGFP moiety next to the SP is indicated (for details, see Section 4). Phyt cleavage sites, as determined in Figure 1 and Figure 2, are shown with red arrows. The ER retrieval signal is marked with the red bottom bracket. The blue bottom bracket is the position of the epitope at the boundary of the N-domain (hatched) and P-domain recognized by the generated anti-CRT3 antibodies (see below). (**B**,**C**) Confocal fluorescence microscopy images of *N. benthamiana* leaf epidermal cells transiently producing EGFP_CRT3_wt (**B**) or EGFP_CRT3_ΔC6 (**C**) proteins. A characteristic reticular localization of the wild-type protein shifted to the more peripheral localization for the truncated one. Images were obtained 2 days post infiltration (dpi) with agrobacteria carrying the respective plasmid. Equal exposure conditions were used in (**B**,**C**). (**D**) Western blot analysis with anti-EGFP antibody of the intracellular (ICF) and apoplastic (Ap) protein fractions obtained from equal weight amounts of leaves transiently producing EGFP_CRT3_wt or EGFP_CRT3_ΔC6, as indicated. Protein fractionation was performed at 2 dpi. Proteins were separated with 12% SDS-gel electrophoresis. M = MW protein markers.

**Figure 4 ijms-24-16527-f004:**
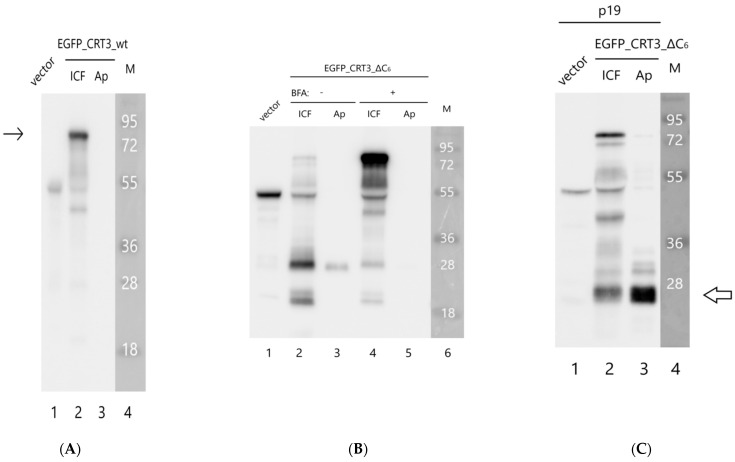
Upon release from the ER, the EGFP_CRT3_ΔC6 protein is processed to generate a 26 kDa fragment capable of secretion into the apoplast. Western blot analyses with anti-CRT3 antibodies of the intracellular (ICF) and apoplastic (Ap) protein fractions from *N. benthamiana* leaves producing EGFP_CRT3_wt (**A**) or EGFP_CRT3_ΔC6 (**B**,**C**). Proteins were separated with 12% SDS-gel electrophoresis. Whereas the EGFP_CRT3_wt protein was largely intact and intracellular (arrow in (**A**)), the full-length EGFP_CRT3_ΔC6 protein was barely detectable even on overexposed blots due to its extensive degradation ((**B**), lane 2). The major ~26 kDa CRT3 fragment thus formed was capable of being exported from the cell (lane 3 in (**B**,**C**)). Treatment of EGFP_CRT3_ΔC6-producing leaves with BFA (15 μg/mL) applied 1 dpi for 24 h markedly suppressed fragmentation of EGFP_CRT3_ΔC6 and secretion of the 26 kDa protein fragment (compare lanes 2 and 3 with lanes 4 and 5 in (**B**)). (**C**) To obtain sufficient material for identification of the 26 kDa CRT3 fragment, EGFP_CRT3_ΔC6 protein was produced in *N. benthamiana* leaves in the presence of the p19 suppressor of silencing for 3 dpi. From the apoplastic fraction, a gel band with electrophoretic mobility of the 26 kDa CRT3 fragment was excised (arrow in (**C**)), and its protein content was characterized by MS analysis (see Figure 5). In (**A**–**C**), vector = total protein from leaves infiltrated with agrobacteria carrying the empty vector. Note that the intensities of the endogenous CRT3 bands (~50 kDa) in these lanes allow evaluation of the relative exposure of each blot. M = MW protein markers. Data were reproducible over three independent experiments.

**Figure 5 ijms-24-16527-f005:**
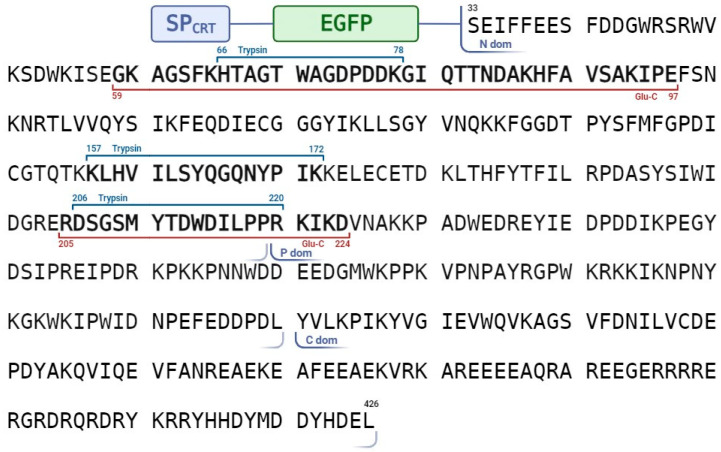
The N-terminal domain of CRT3 is secreted into the apoplast. Sequence coverage of the apoplastic 26 kDa CRT3 fragment originated from EGFP_CRT3_ΔC6 protein with mass spectrometry-identified peptides. Sequences of tryptic (blue upper bracket) and Glu-C (red bottom bracket) peptides for which identity was confirmed by MS/MS fragmentation are shown. Boundaries of the N-, P-, and C-terminal domains of *Nt*CRT3 are indicated below the amino acid sequence.

## Data Availability

Data are contained within the article and Appendix A.

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
