# Peer review of "Phytaspase Is Capable of Detaching the Endoplasmic Reticulum Retrieval Signal from Tobacco Calreticulin-3"

_ijms, 2023, doi:10.3390/ijms242216527_

Round 1

Reviewer 1 Report

Comments and Suggestions for Authors

Dear Authors,

I have an opportunity to review manuscript entitled: “Phytaspase is Capable of Detaching the Endoplasmic Reticulum Retrieval Signal from Tobacco Calreticulin-3” submitted to IJMS;

Authors concentrated on Nicotiana tabacum phytaspase, a plant aspartate-specific protease, introduces two breaks at the C-terminus of the N. tabacum ER resident calreticulin-3. These cleavages resulted in removal of either a dipeptide or a hexapeptide from the 22 C-terminus of calreticulin-3 encompassing part or all of the ER retrieval signal. What is important, these cleavages resulted in removal of either a dipeptide or a hexapeptide from the C-terminus of calreticulin-3 encompassing part or all of the ER retrieval signal. Consistently, expression of the calreticulin-3 derivative mimicking the phytaspase cleavage product in Nicotiana benthamiana cells demonstrated loss of the ER accumulation of the protein.

Introduction gives the reader firm but sufficient background to analysis results. Between Introduction part and materials and methods Authors added the most important demonstrated facts coming from results. I suggest to add also precise aim of the studies;

Materials and methods are precisely described;

Results are clearly described. Supplementary data are informative  especially figure S3, please, rethink adding this figure to the main manuscript;

Figure 3 – western blott visualization is weak- please, work on it as far as possible;

In conclusion, Authors stated that obtained results can be “ implications further fate of the escaped resident”, therefore I suggest to add future prospects which can make the results more visible to the wider audience;

Author Response

We would like to thank the reviewers for their time and helpful comments. Below we address each of the comments one by one.

Reviewer 1 Comments for the Author

 Q1. Introduction gives the reader firm but sufficient background to analysis results. Between Introduction part and materials and methods Authors added the most important demonstrated facts coming from results. I suggest to add also precise aim of the studies.

Answer: Done; please see p. 2 in the revised version of the manuscript.

Q2. Results are clearly described. Supplementary data are informative  especially figure S3, please, rethink adding this figure to the main manuscript.

Answer: Done. We have moved Figure S3 from the Supplement to the main text (Figure 5 in the revised version of the manuscript). 

Q3. Figure 3 – western blot visualization is weak- please, work on it as far as possible;

Answer: Done. We improved quality of Figure 3D, please see Figure 3D in the revised version of the manuscript.

Q4.  In conclusion, Authors stated that obtained results can be “implications further fate of the escaped resident”, therefore I suggest to add future prospects which can make the results more visible to the wider audience.

Answer: Our phrase ‘with implications for the further fate of the escaped resident’ is from the Abstract. We agree with the reviewer that some additional detalization would be helpful, yet this is hardly possible due to the space limit of the Abstract. We would therefore suggest to keep the perspectives (conditions triggering CRT3 processing; involvement of the secreted or re-imported phytaspase; apoplastic CRT3 fragment as a signaling molecule) in the Discussion section.

Reviewer 2 Report

Comments and Suggestions for Authors

This manuscript describes the in vitro activity of a plant protease in cleavage of a C-terminal ER retrieval signal from the ER lectin chaperone calreticulin 3. While overall this manuscript is well written and the data and conclusions are clear, I miss important information on the biological significance of the findings. In particular, I would like to see additional data showing that the observed cleavage also occurs in vivo in plants and could have a biological function.

 Specific comments:

The analysed CRT3 from N. tabacum was produced in E. coli. I wonder whether the authors did any experiments to show that the recombinant CRT3 produced in bacteria is proper folded and functional. Plant CRT3 has posttranslational modifications that are not made in the same way in bacteria and it is possible that the used CRT3 is misfolded and that the analysed phytaspase acts only on misfolded CRT3.

 To show that the cleavage occurs in vivo the authors could, for example, transiently co-express the GFP-tagged NtCRT with the phytaspase and see if it also gets secreted. The authors could also express the phytaspase and analyse by immunoblotting if the processed 26 kDa CRT3 fragment is detectable in the apoplast which would also indicate that the protease could in vivo process CRT3.

 How specific is the cleavage? Did you also observed minor amounts of cleavage product cleaved at D421 or D417?

Author Response

We would like to thank the reviewers for their time and helpful comments. Below we address each of the comments one by one.

Reviewer 2 Comments for the Author

Q1. This manuscript describes the in vitro activity of a plant protease in cleavage of a C-terminal ER retrieval signal from the ER lectin chaperone calreticulin 3. While overall this manuscript is well written and the data and conclusions are clear, I miss important information on the biological significance of the findings. In particular, I would like to see additional data showing that the observed cleavage also occurs in vivo in plants and could have a biological function.

Answer: The main aim of the present study was to demonstrate, for the first time, that a proteolytic enzyme capable of detaching the ER retrieval signal from a soluble ER resident protein, a hypothetical possibility that has been discussed for decades (see, for example, refs. 15-18,20), does exist. We agree with the reviewer that identification of conditions that could trigger processing of this kind now becomes of utmost importance, and we put this as the main task for the future (see Discussion).

 Q2. Specific comments:

The analysed CRT3 from N. tabacum was produced in E. coli. I wonder whether the authors did any experiments to show that the recombinant CRT3 produced in bacteria is proper folded and functional. Plant CRT3 has posttranslational modifications that are not made in the same way in bacteria and it is possible that the used CRT3 is misfolded and that the analysed phytaspase acts only on misfolded CRT3.

Answer: Our previously published data that served as a starting point for the current study documented an analogous shift in the electrophoretic mobility of CRT3 isolated from phytaspase-producing N. benthamiana leaves in BioID experiments (ref. 36). We therefore consider highly likely that CRT3 in its natural environment is sensitive to phytaspase-mediated cleavage as well.

 Q3. To show that the cleavage occurs in vivo the authors could, for example, transiently co-express the GFP-tagged NtCRT with the phytaspase and see if it also gets secreted. The authors could also express the phytaspase and analyse by immunoblotting if the processed 26 kDa CRT3 fragment is detectable in the apoplast which would also indicate that the protease could in vivo process CRT3.

Answer: We are grateful to the reviewer for these suggestions. Actually, we performed these experiments but failed to obtain a definitive answer thus far. The main problem is that the effects observed in plants are only partial and hence less explicit. Also, it appears that the increase in the level of phytaspase per se may be insufficient to induce processing of CRT3. In our BioID experiments, for example, leaves were supplied with ATP, biotin, magnesium acetate to promote protein biotinylation (ref. 36). We do not know whether any of these (or other) components could contribute to processing of CRT3. To our mind, an important message from our study is that such conditions are worth searching, although this may be not an easy task.

Q4. How specific is the cleavage? Did you also observed minor amounts of cleavage product cleaved at D421 or D417?

Answer: Hydrolysis at D417 or D421 should produce semi-LysC peptides with m/z 883.4 and 1407.6, respectively. Careful inspection of MS spectra demonstrated complete absence of these peptides (see also Figure S1), thus confirming the exceptional specificity of phytaspase. Perhaps this is not surprising, as phytaspase recognizes a tetrapeptide motif in its substrates (see, for example, refs. 26-29,32,33), rather than simply the Asp residue.

Round 2

Reviewer 2 Report

Comments and Suggestions for Authors

The authors did not provide any further experimental evidence showing that this processing is indeed a natural process with biological significance. I agree, however, that this task is quite challenging because the process could be triggered by certain environmental conditions that are unknown.